# Mechanistic Exploration of Visible Light-Activated Carbon/TiO₂ Hybrid Dots Damaging Bacterial Cells

Audrey F. Adcock [1], Weixiong Liang [2], Peter A. Okonjo [1], Xiuli Dong [1], Kirkland Sheriff [2], Ping Wang [2], Isaiah S. Ferguson [1], Shiou-Jyh Hwu [2], Ya-Ping Sun [2,*] and Liju Yang [1,*]

[1] Department of Pharmaceutical Sciences, Biomanufacturing Research Institute and Technology Enterprise (BRITE), North Carolina Central University, Durham, NC 27707, USA
[2] Department of Chemistry, Clemson University, Clemson, SC 29634, USA
* Correspondence: syaping@clemson.edu (Y.-P.S.); lyang@nccu.edu (L.Y.)

**Abstract:** The carbon/TiO₂ hybrid dots (C/TiO₂-Dots) are structurally TiO₂ nanoparticles (in the order of 25 nm in diameter from commercially available colloidal TiO₂ samples) surface-attached by nanoscale carbon domains with organic moieties, thus equivalent to hybrids of individual TiO₂ nanoparticles each decorated with many carbon dots. These hybrid dots with exposure to visible light exhibit potent antibacterial properties, similar to those found in neat carbon dots with the same light activation. The results from the use of established scavengers for reactive oxygen species (ROS) to "quench" the antibacterial activities, an indication for shared mechanistic origins, are also similar. The findings in experiments on probing biological consequences of the antibacterial action suggest that the visible light-activated C/TiO₂-Dots cause significant damage to the bacterial cell membrane, resulting in higher permeability, with the associated oxidative stress leading to lipid peroxidation, inhibiting bacterial growth. The induced bacterial cell damage could be observed more directly in the transmission electron microscopy (TEM) imaging. Opportunities for the further development of the hybrid dots platform for a variety of antibacterial applications are discussed.

**Keywords:** carbon/TiO₂; hybrid dots; light-activated; bacteria; antimicrobial; mechanism

## 1. Introduction

Carbon "quantum" dots or carbon dots (CDots) may be defined generally as small carbon nanoparticles with various surface passivation schemes [1–3], which may be considered loosely as core-shell nanostructures, each with a carbon nanoparticle core and a thin shell of soft organic materials (Figure 1) [3]. There has been widespread interest in CDots and related carbon/organic nanostructures, with the emergence of a rapidly advancing and expanding research field [3–16]. Among actively pursued technological applications of CDots are those exploiting their visible/natural light-activated antimicrobial properties [15], with many experimental demonstrations on the effective inactivation of various microorganisms, including multidrug-resistant bacterial pathogens, and viruses [17–21].

Low-dimensional carbon materials have historically been employed in various uses that combine the carbon with metals and/or metal oxides, for which a good example is the application in some popular catalysts. In the more recent development of the different nanoscale carbon allotropes, from carbon nanotubes and graphene nanosheets to carbon nanoparticles, there has been significant experimental evidence suggesting structural compatibility of the nanoscale carbon with other popular nanomaterials, including especially conventional nanoscale semiconductors, for the preparation of hybrid nanostructures [22–25]. For small carbon nanoparticles, which represent zero-dimensional carbon allotrope, their ready coating by selected metals in simple photoreductive deposition was demonstrated [26,27], with the corresponding hybrid CDots serving as highly potent photocatalysts [26]. Additionally demonstrated was the similarly successful coating of small

carbon nanoparticles with conventional semiconductors such as ZnS and ZnO for their derived hybrid CDots to exhibit much brighter fluorescence emissions [28].

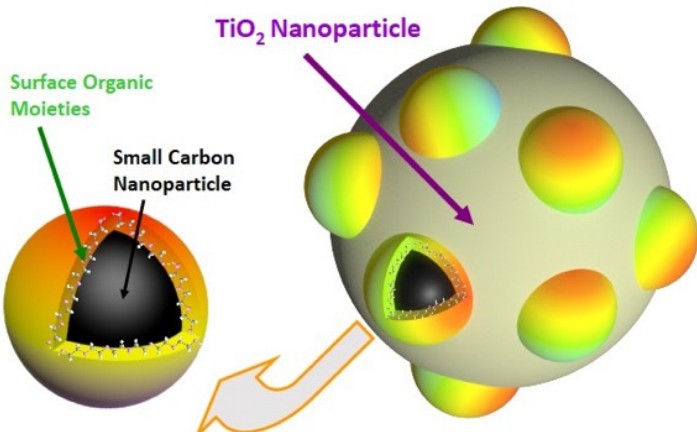

**Figure 1.** Cartoon illustration on C/TiO$_2$ hybrid dots (**right**), with the entities on the TiO$_2$ nanoparticle surface each being a nanoscale carbon domain with organic moieties, thus equivalent to the classically defined CDots (**left**) as small carbon nanoparticles with the surface passivation by organic functionalization.

Colloidal TiO$_2$ has been used extensively for a variety of purposes, including applications exploiting the light-activated antimicrobial function [29]. However, the required excitation of TiO$_2$ nanoparticles by hazardous UV light represents a serious limitation. Hybrid nanostructures containing both TiO$_2$ and carbon have attracted recent attention, taking advantage of the ability of the carbon domains in the nanostructures to harvest photons effectively over the visible spectrum. For example, hybrid nanostructures of small or larger TiO$_2$ nanoparticles combined with carbon and organic functionalization were prepared [18,25,30], and such nanostructures, denoted as C/TiO$_2$ hybrid dots, were evaluated for their antimicrobial properties [18,30]. Interestingly, the C/TiO$_2$ hybrid dots derived from the commercially available larger TiO$_2$ nanoparticles (25–30 nm in diameter, Figure 1) were found to be much more effective than those from the specifically synthesized small TiO$_2$ nanoparticles (8 nm in diameter, for example) in the visible light-driven antibacterial activities [18]. The valuable implication of the finding is such that the larger TiO$_2$ nanoparticles are not only available commercially in extremely large quantities but also very inexpensive, making their derived hybrid dots particularly promising for potentially widespread antimicrobial applications.

In this study, we further investigated our previously reported C/TiO$_2$ hybrid dots (Figure 1) for their visible light-activated antibacterial activities and related mechanistic implications, with particular emphasis on the examination and understanding of the biological consequences to the targeted bacterial cells in terms of major damage to the cell structures and functions. Additionally presented and discussed are opportunities and challenges in the development of carbon-based/derived hybrid dots for antimicrobial applications.

## 2. Experimental Section

The preparation and characterization of the C/TiO$_2$ hybrid dots were largely the same as those reported previously [18], with the details provided as Supplementary Materials. The antibacterial experiments on these dots followed the protocols developed and used for other dot samples in previously reported studies, which are also provided as Supplementary Materials.

### 2.1. Intracellular ROS Measurement

For measurement of intracellular ROS in bacterial cells upon the C/TiO$_2$ hybrid dots treatment, the cells were treated with the hybrid dots under visible light for 2 h, the cells were then collected in 1.5 mL centrifuge tubes, followed by centrifugation at 8000× *g* for

10 min and removal of the supernatant. Aliquot of 320 μL of 10 μM dihydrorhodamine 123 (DHR123) was then added to each tube. DHR 123 is a probe widely used to detect several reactive species. DHR is oxidized to rhodamine 123 which is highly fluorescent around 536 nm with excitation at 500 nm. After the addition of DHR 123, the tubes were vigorously vortexed and incubated at the room temperature for 40 min. After centrifugation, discard of supernatant, and a rinse with 0.85% NaCl, the cells were resuspended in 320 μL 0.85% NaCl solution. The fluorescence (ex/em, 500/535 nm) of each sample was measured using a SpectraMax M5 microplate reader. With the fluorescence intensity at 535 nm of the controls as the baseline, the fold increase in fluorescence intensity in the dot-treated samples was used as the measurement for the fold increase in intracellular ROS generation induced by the dot treatments.

### 2.2. Membrane Damage—Live/Dead Bacterial Staining

To examine bacterial membrane damage after dot treatment, the live/dead bacterial staining assay was performed using Live/Dead Baclight bacterial viability kit (Invitrogen, Eugene, OR). The kit employs two nucleic acid dyes: the green SYTO 9 and the red propidium iodide (PI) dye. The PI dye penetrates only bacteria with damaged membranes, so the kit stains the live cells with intact membranes in green, and the dead cells with damaged membranes in red. The staining process was carried out according to the manufacturer's manual. After staining, the fluorescent images of the stained bacterial cells were taken on an Olympus IX51fluorescence microscope.

### 2.3. Measurement of Lipid Peroxidation

*B. subtilis* cells (~$10^7$ CFU/mL) were treated with 0.1 mg/mL PEI&PEG–C/TiO$_2$-Dots and visible light for 2 h using the same protocol as described above. The treated cell samples and the controls were collected, centrifuged at 8000× *g* for 10 min, and then resuspended in 200 μL of PBS buffer. Then, 2 μL of 100× BHT reagent was added to the samples and controls. Lipid peroxidation levels in the treated cells and controls were measured by quantifying malondialdehyde (MDA) using a commercial kit (Oxiselect™ TBARS Assay Kit, Fisher Scientific, Fair Lawn, NJ, USA). Briefly, 100 μL of the bacterial samples and the standard reactions were transferred to new microfuge tubes followed by the addition of 100 μL of SDS lysis solution, incubated for 5 min at room temperature. Then, 250 μL of TBA reagent was added and incubated at 95 °C for 45–60 min. The tubes were cooled down for 5 min in ice bath, and then centrifuged at 3000× *g* for 15 min. The supernatant was collected for measurement by fluorescence. For fluorescence measurement, 150 μL of the samples were dispensed into the wells of the 96 well Costar plate, and fluorescence intensity at 590 nm for each sample was measured with excitation at 540 nm. The MDA standard curve was established using the MDA standards in the kit according to manufacturer's instructions, and was used for determining the levels of MDA in the treated samples and the controls.

### 2.4. Transmission Electron Microscopy (TEM) Imaging

*B. subtilis* cells were treated with the hybrid dots at 0.1 mg/mL for 2 h under light illumination. After treatment, 600 μL of both the treated and control samples were fixed with the fixative chemical (4% glutaraldehyde in 0.1 M sodium cacodylate buffer, pH 7.4). The contents were mixed by gently inverting the tube back and forth. Centrifugation was done at 1000× *g* for 10 min. The samples were stored overnight at 4 °C prior to the next procedure. The pellet was rinsed in 0.1 M sodium cacodylate buffer several times, and fixed in 1% buffered osmium tetroxide for one hour at room temperature. Following dehydration with graded series of ethanol (30%, 50%, 75%, 100%, 100%) and two changes of propylene oxide, the cells were infiltrated and embedded in PolyBed 812 epoxy resin (Polysciences, Inc., Warrington, PA, USA). Ultrathin sections (70 nm) were cut and mounted on copper grids followed by staining with 4% uranyl acetate and 0.4% lead acetate. Sections were imaged under a JEOL JEM-1230 transmission electron microscope (JEOL USA, Inc., Peabody, MA,

USA) operating at 80 kV, with digital images acquired with a Gatan Orius SC1000CCD Digital Camera and Gatan Microscopy Suite v3.0 software (Gatan, Inc., Pleasanton, CA, USA).

## 3. Results and Discussion

The C/TiO$_2$ hybrid dots (Figure 1) were prepared by carbonizing organic precursor molecules on the surface of existing TiO$_2$ nanoparticles, with the remaining or residual organic molecular segments that survived the carbonization processing conditions serving the surface passivation function [18]. For the preparation, the commercially acquired colloidal TiO$_2$ sample (Degussa P25) was fractionated to harvest the more aqueous dispersible nanoparticles of sizes around 25–30 nm in diameter. The resulting TiO$_2$ nanoparticles in aqueous suspension were mixed well with the selected organic molecules, followed by the solvent removal to obtain a solid-state mixture for the microwave-assisted thermal carbonization processing, as reported previously [18]. The selected precursor organic molecules were oligomeric polyethylene glycol (PEG, average molecular weight ~1500) and its mixture with oligomeric polyethylenimine (PEI, average molecular weight ~600). Their corresponding hybrid dots are denoted as PEG-C/TiO$_2$-Dots and PEI&PEG-C/TiO$_2$-Dots, respectively [18].

In the preparation of the hybrid dots, the thermal processing conditions are unlikely to cause any meaningful changes to the TiO$_2$ nanoparticles. According to X-ray powder diffraction results (Figure 2), the hybrid dot sample contains largely amorphous carbon in addition to the TiO$_2$ nanoparticles. One might argue that the sample could be a simple mixture of the TiO$_2$ nanoparticles with CDot-like entities produced in the thermal processing, but countering such an argument is that the sample was cleaned via prolonged dialysis in the membrane tubing of pore-size 25,000 MWCO. For the control sample prepared without the TiO$_2$ nanoparticles, the same cleaning procedure with the dialysis eliminated essentially the entire sample.

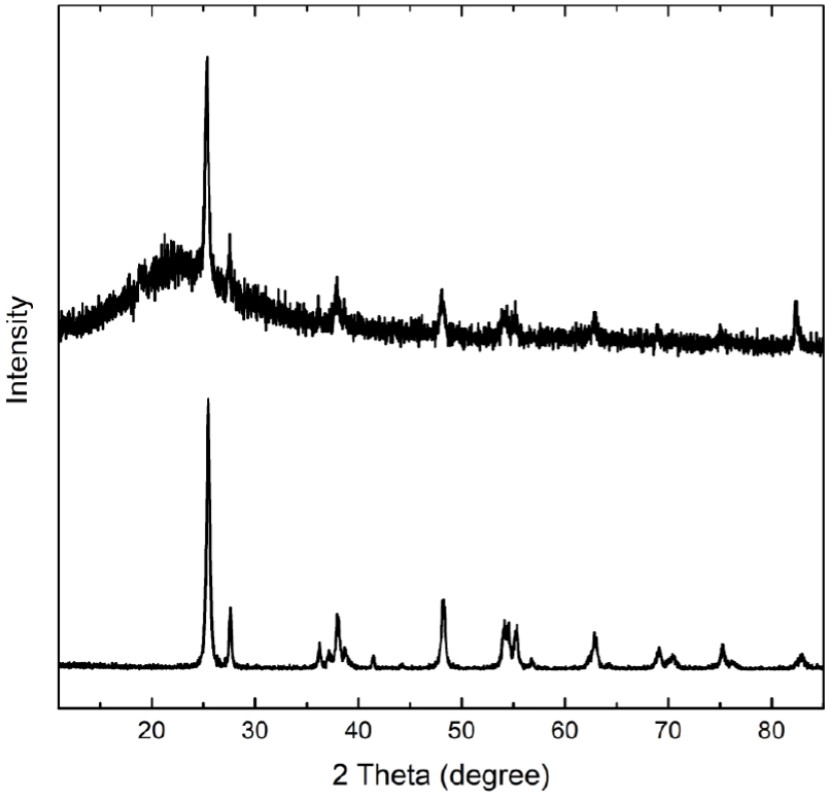

**Figure 2.** Powder X-ray diffraction patterns of C/TiO$_2$-Dots (upper) and the precursor TiO$_2$ nanoparticles (lower) from the commercially acquired Degussa sample.

Colloidal $TiO_2$ is crystalline, with high microwave cross-sections, thus expected to absorb the microwave energy preferably in the microwave-assisted thermal processing to create very hot $TiO_2$ nanoparticles for the carbonization of the organic species in the vicinity. Such an experimental configuration for the thermal carbonization of organic species mostly on the very hot $TiO_2$ nanoparticle surface should be favorable to the formation of the targeted $C/TiO_2$ hybrid dot structure (Figure 1), namely individual $TiO_2$ nanoparticles each surface-attached/decorated with nanoscale carbon domains and residual organic species. The latter are structurally equivalent to carbon nanoparticles with surface organic functionalization, or CDot-like, making the hybrid dot structurally analogous to a $TiO_2$ nanoparticle bound with many CDots. Thus, the use of microwave irradiation to deliver the energy required for thermal carbonization is ideally suited for the targeted hybrid dot structure (Figure 1).

$C/TiO_2$-Dots due to the hydrophilic organic species on the surface are readily soluble in aqueous or highly polar organic media. In these dots, the $TiO_2$ nanoparticles are known to absorb only in the UV (cut-off shorter than 400 nm for aqueous suspended $TiO_2$ nanoparticles of 25–30 nm in diameter), so that the observed optical absorptions of $C/TiO_2$-Dots over the visible spectral region must be due entirely to the CDot-like organic functionalized nanoscale carbon domains (Figure 1), comparable with the optical absorptions of the corresponding neat CDots without any $TiO_2$ and also those of aqueous suspended small carbon nanoparticles (Figure 3). The fluorescence spectra of the $C/TiO_2$-Dots are also similar to those of their neat counterparts containing no $TiO_2$ (Figure 3).

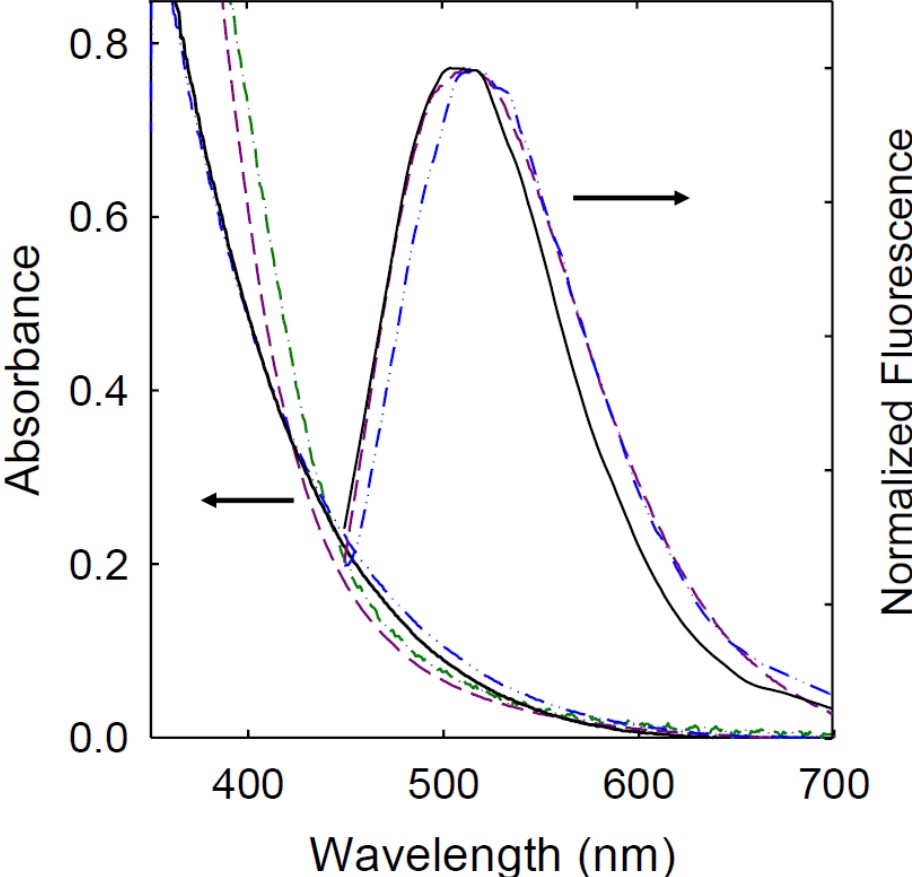

**Figure 3.** Absorption and fluorescence spectra of PEI&PEG-$C/TiO_2$-Dots (solid, black) and PEG-$C/TiO_2$-Dots (dash-dot-dot, blue), with those of PEI-CDots (dash, purple) and aqueous suspended small carbon nanoparticles (dash-dot, green, absorption only) for comparison.

Both PEG-$C/TiO_2$-Dots and PEI&PEG-$C/TiO_2$-Dots with exposure to visible light were found to kill bacterial cells, as reported previously [18]. With each hybrid dot consid-

ered as structurally analogous to a TiO$_2$ nanoparticle bound with many CDot-like entities, the observed antibacterial activities may be contributed by both TiO$_2$ and CDot-like components in the dot structure. When exposed to visible light, the CDot-like domain is responsible for the photon harvesting, with the harvested excitation energy driving its own antibacterial action and also activating the attached TiO$_2$ nanoparticle in a scheme rather similar to the known photosensitization in dye-modified colloidal TiO$_2$ [31–33]. As related, the observed meaningfully lower fluorescence quantum yields of the hybrid dots than those of their neat counterparts could be attributed to the intra-dot quenching effects due to electron transfers [18], again similar to those known in dye-sensitized TiO$_2$ systems [31–33].

Mechanistically on neat (no TiO$_2$) CDots, the photoexcitation drives rapid charge transfers and separation for electrons and holes. The separated redox pairs are trapped at the likely abundant surface defect sites of the carbon nanoparticles, which in the CDots are stabilized by the organic functionalization [3]. The available experimental results from the ultrafast transient absorption probing [34,35] and the photoinduced reductive noble metal deposition on the dot surface [26,27] are consistent with the generation of the redox pairs following photoexcitation. The radiative recombination of the separated redox pairs results in longer-lived emissive excited states (in nanoseconds as determined experimentally) responsible for the observed bright and colorful fluorescence emissions [3,36]. On the light-activated antimicrobial function of CDots, the mechanistic rationale based on available experimental results called for combined actions of the two sets of highly reactive species, the initially formed electron–hole pairs and the classical reactive oxygen species (ROS) generated in the emissive excited states [37]. In the same mechanistic framework, the TiO$_2$ nanoparticles in the hybrid dots act as a quencher of the photoexcited CDot-like entities and by extension their antimicrobial actions, but at the same time the sensitization due to the quenching activates the antimicrobial function of the TiO$_2$ nanoparticles. An interesting question is whether the sensitization of the TiO$_2$ nanoparticles is associated with the quenching of the initially formed redox pairs or the longer-lived emissive excited states or both, though it is beyond the scope of this work and best left for a dedicated investigation. Nevertheless, the results of ROS scavenging experiments discussed as follows support the notion that the observed antimicrobial properties of the C/TiO$_2$ hybrid dots, similar to those of their neat counterparts, are beyond just the classical ROS.

Experimentally for the detection and quantification of the intracellular ROS levels in the treatment of *Bacillus subtilis* cells with the hybrid dots and visible light, dihydrorhodamine (DHR 123) was used as a probe [38,39]. DHR 123 can be oxidized by ROS to form rhodamine 123, which with 500 nm excitation is brightly fluorescent around 536 nm [38,39]. Shown in Figure 4A is the significant increase in intracellular ROS generated in *B. subtilis* cells after their being treated with the hybrid dots and visible light illumination. For the treatment with 0.1 mg/mL PEG&PEI-C/TiO$_2$-Dots and visible light for 2 h, the intracellular ROS level in the cells is about 6 times of that in the untreated cells (Figure 4A).

In a subsequent experiment, we examined how the traditional ROS scavengers, L-histidine and *t*-butanol, could affect the antimicrobial outcome of C/TiO$_2$ hybrid dot treatments. Generally, L-histidine and *t*-butanol are known as being able to scavenge singlet molecular oxygen and hydroxyl radicals, respectively. Compared in Figure 4B are the viable cell number changes in *B. subtilis* cells due to the treatment of PEG&PEI-C/TiO$_2$-Dots with visible light in the absence and presence of L-histidine. The treatment with the dot concentration of 0.1 mg/mL under visible light resulted in ~4.6 log reduction in viable cells in the absence of L-histidine, but only ~1.9 log reduction in the presence of L-histidine, consistent with the known protective effect of L-histidine as an ROS scavenger. However, the protection by L-histidine was limited, reducing the magnitude of log reduction in viable cells by ~60% only even at the high L-histidine concentration of 60 mM (near the solubility limit), far from being complete.

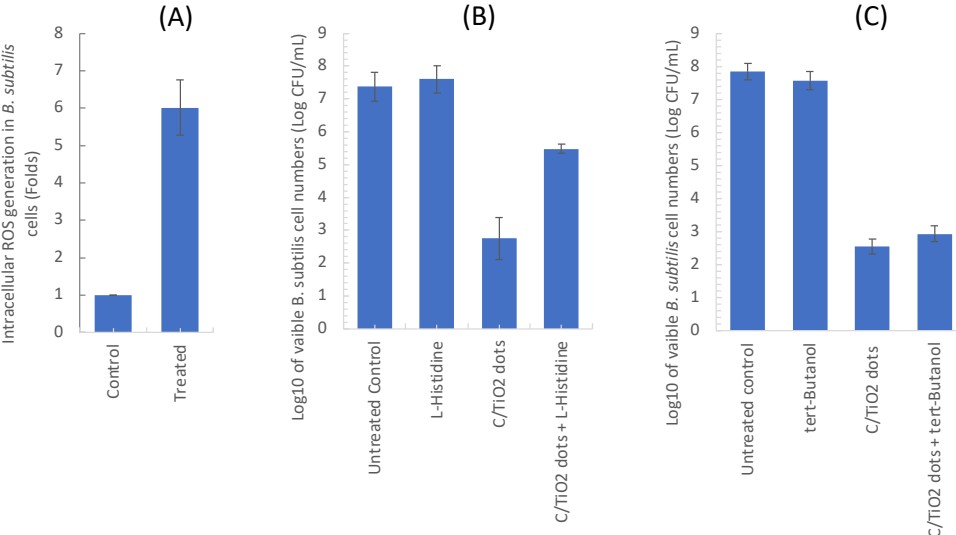

**Figure 4.** (**A**) The fold increase in intracellular ROS generation; and (**B**) the viable cell number change in *B. subtilis* after the cells were treated with C/TiO$_2$ hybrid dots in the presence and absence of ROS scavenger L-histidine, and (**C**) in the absence and presence of ROS scavenger *t*-butanol, along with those of untreated control cells. Treatment: 0.1 mg/mL PEI&PEG-C/TiO$_2$-Dots with visible light illumination for 2 h. Scavenger concentration: 60 mM.

The same experiments with *t*-butanol as ROS scavenger were performed, with the results showing no protective effect even at the high *t*-butanol concentration of 60 mM. The outcomes for both L-histidine and *t*-butanol are essentially the same as those found in similar experimental evaluations on the protective effects of these scavengers in the treatment of bacterial cells with neat CDots and visible light exposure, namely some limited protection by L-histidine and none by *t*-butanol [37]. In the mechanistic framework that the antibacterial activities of light-activated CDots are due to the combined actions of two sets of highly reactive species, the initially formed electron–hole pairs and the classical ROS generated in the emissive excited states [37], the different "lifetimes of killing" between the two sets of photo-generated reactive species in CDots could be used to rationalize the observed inadequate protection by the scavengers, such that L-histidine is only capable of quenching the longer lived set of reactive species (the classical ROS generated in the emissive excited states, whose lifetimes are in nanoseconds [40]). Interestingly, the scavenging outcomes were not affected in any fundamental fashion by the presence of colloidal TiO$_2$ in the hybrid dots, suggesting the preservation of the basic photoexcited state processes and the actions of the different reactive species on different time scales in the structurally and composition-wise more complex hybrid dots. However, the likely "sharing" of the excitation energy between the initially excited nanoscale carbon domains (neat CDot-like, Figure 1) and the photo-sensitized TiO$_2$ nanoparticles in the hybrid dots, both contributing to the observed antimicrobial activities, represents added complications mechanistically. Important questions such as the time scale of the photo-sensitization and potentially differential quenchings of the reactive species at the carbon and TiO$_2$ sides by the scavengers remain to be addressed in further dedicated investigations.

The obvious major difference between the two scavengers L-histidine and *t*-butanol is puzzling, beyond the mechanistic rationale discussed above. However, it might be speculated that the difference could be due to different lifetimes of their targeted classical ROS, namely that the hydroxyl radicals in aqueous medium are too short lived for *t*-butanol to scavenger.

The consequences of the treatment with the C/TiO$_2$ hybrid dots and visible light include the suppression of the intrinsic antioxidant defense capacity of the cells, causing various adverse effects and damage to the cells, such as the oxidation of proteins and

the disruption of cell membranes [41,42]. Often accompanying cell membrane damage is the lipid peroxidation [43], a multiple-step process in which lipid carbon-carbon double bonds are attacked by free radicals and/or strong oxidants [44–46]. Some studies have demonstrated that free radicals can induce cell membrane damage [47], and the oxidative stress can lead to lipid peroxidation, inhibiting bacterial growth [48]. Other studies have found that the membrane damage by photoexcited $TiO_2$ is often associated with lipid peroxidation [49]. The lipid peroxidation produces lipid hydroperoxides (LOOH) and various aldehydes including especially malondialdehyde (MDA) [50], which is commonly used as an indicator of lipid peroxidation. In this study, the extent of lipid peroxidation in *B. subtilis* cells upon the treatments of $C/TiO_2$-Dots with visible light was measured by quantifying MDA with a commercial kit (TBARS Assay), which is based on the reaction of MDA with thiobarbituric acid (TBA) to form a red fluorescent adduct [51,52]. Shown in Figure 5A is the increased amount of MDA measured in *Bacillus* cells after the treatment with 0.1 mg/mL PEI&PEG-$C/TiO_2$-Dots and visible light for 2 h. The amount is significantly higher than that in the untreated control samples, indicating the cell membrane lipid peroxidation caused by the highly reactive species produced in the treatment.

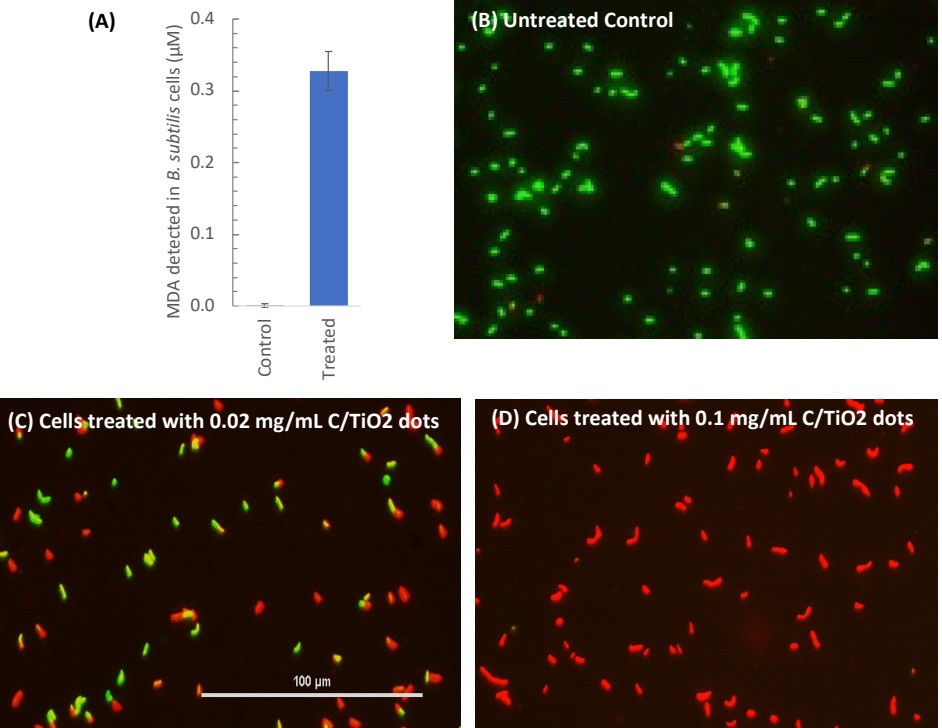

**Figure 5.** (**A**) The increased amount of MDA in *B. subtilis* cells after they were treated with PEI&PEG-$C/TiO_2$-Dots at 0.1 mg/mL with visible light for 2 h. (**B–D**) The fluorescent images of *B. subtilis* cells without dots treatment, with 0.02 mg/mL, and 0.1 mg/mL PEG&PEI-$C/TiO_2$-Dots treatment with visible light for 2 h, respectively, and stained with live/dead bacterial viability kit.

The damage to the cell membrane can be assessed more directly with the live/dead bacterial viability kit, which contains two nucleic acid dyes, SYTO 9 and propidium iodide (PI), to stain the live and dead cells for their significant differences in the membrane permeability. The green SYTO 9 dye can penetrate membranes of both live and dead cells, but the red PI dye can only penetrate the damaged cell membranes of dead cells. Thus, the kit stains the live cells with intact membranes in green, and the dead cells with damaged membranes in red. Figure 5B–D illustrates the changes in cell permeability due to the treatment with PEG&PEI-$C/TiO_2$-Dots and visible light. In control samples, almost all cells were viable and stained green (Figure 5B), but in the treated cells there was obvious damage to the cell membrane, more evident when higher dot concentrations were used

in the treatments, as demonstrated by the increase in the number of red-stained cells (Figure 5C,D). The live/dead staining results clearly confirmed that the treatments with C/TiO$_2$ hybrid dots damaged bacterial membrane to result in higher permeability.

The cells post-treatment with the hybrid dots were also examined by TEM imaging. Shown in Figure 6 are TEM images of untreated control cells and the treated cells. For the untreated control cells, their structures show regularly situated genomic materials surrounded by the cytoplasmic area, with clearly organized cell membrane and wall and the periplasmic spaces. For the treated cells, there is shrinkage of the cytoplasm, with a separation of the cytoplasm from the cell wall, and there are also some distortions and collapse in the membrane-wall continuity, most likely due to the cell membrane damage. The TEM imaging results also suggest that inside the treated cells, the genomic area is disorganized, with the distinct condensation of cytoplasmic materials probably resulted from the degradation of proteins or other intracellular components induced by the hybrid dots treatment. These alterations in treated cells are similar to those found previously in cells treated by neat CDots with visible light [37]. The similarity is consistent with the results and discussion above on the C/TiO$_2$ hybrid dots sharing the same mechanistic framework with neat CDots for their highly potent photo-activated antimicrobial functions.

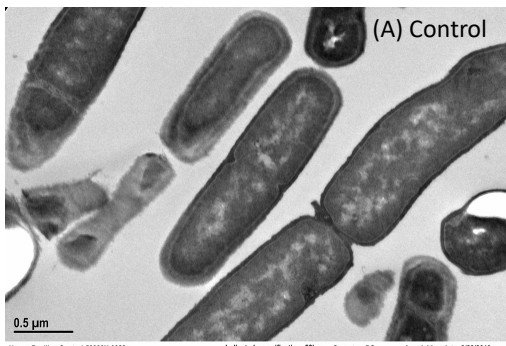 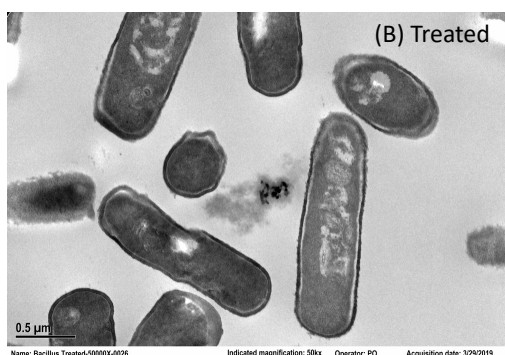

**Figure 6.** Representative TEM images of *B. subtilis* cells: (**A**) untreated, and (**B**) after treated with 0.1 mg/mL PEI&PEG-C/TiO$_2$-Dots with visible light for 2 h.

## 4. Summary and Conclusions

Nanoscale carbon is compatible with colloidal TiO$_2$ for C/TiO$_2$ hybrid structures, which with the surface organic moieties are equivalent to individual TiO$_2$ nanoparticles each surface-attached by many CDot-like nanoscale entities, hence C/TiO$_2$ hybrid dots. These hybrid dots are optically absorptive in the visible spectrum, due exclusively to the absorptions of the nanoscale carbon domains. The photon energies thus harvested by the hybrid dots could drive their antibacterial activities, similar to those found in neat CDots (no TiO$_2$) with the same visible light exposure. The mechanistic origins of the antibacterial function shared by the C/TiO$_2$ hybrid dots and their neat counterparts, as supported by the similar outcomes of the experiments on using established scavengers of reactive oxygen species (ROS) to "quench" the antibacterial activities, are also likely to be similar. In experiments for probing biological consequences of the antibacterial action, the results suggest that the visible light-activated C/TiO$_2$ hybrid dots cause significant damage to the bacterial cell membrane to result in higher permeability, and the associated oxidative stress leads to lipid peroxidation, inhibiting bacterial growth. For the bacterial cell damage, more direct evidence is provided by the TEM imaging results. Since the colloidal TiO$_2$ is widely available commercially in extremely large quantities and very low costs, from which C/TiO$_2$-Dots can be produced in a facile fashion, the hybrid dots represent a highly promising platform for a variety of antibacterial applications. Beyond antimicrobial uses, the hybrid dots are more advantageous than the widely pursued dye-sensitized colloidal TiO$_2$ for broad applications, such as in photocatalysis for CO$_2$ conversion and waste management.

**Supplementary Materials:** The following supporting information can be downloaded at: https://www.mdpi.com/article/10.3390/app12199633/s1, Information about Materials, Measurement, Synthesis of C/TiO$_2$ Hybrid Dots, Microscopy results, and Antibacterial Experiments; Figure S1: Representative AFM (upper) and TEM (lower) images of PEI&PEG-C/TiO$_2$-Dots. Inset: Statistical analysis of dot sizes based on multiple AFM images.

**Author Contributions:** A.F.A.: investigation, methodology. W.L.: investigation, methodology, P.A.O.: investigation, methodology. X.D.: data curation, investigation, methodology. K.S.: investigation, methodology. P.W.: data curation, investigation, methodology. I.S.F.: investigation. S.-J.H.: investigation, methodology. Y.-P.S.: conceptualization, funding acquisition, resources, supervision, writing—original draft, writing—review & editing, L.Y.: conceptualization, funding acquisition, investigation, resources, supervision, writing—original draft, writing—review & editing. All authors have read and agreed to the published version of the manuscript.

**Funding:** Financial support from NSF (2102021 and 2102056, and 1855905) and USDA (2019-67018-29689) is gratefully acknowledged.

**Institutional Review Board Statement:** Not applicable.

**Informed Consent Statement:** Not applicable.

**Data Availability Statement:** All data is contained within the article or Supplementary Materials.

**Conflicts of Interest:** The authors declare no conflict of interest.

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
