# Peer review of "Mechanistic Exploration of Visible Light-Activated Carbon/TiO2 Hybrid Dots Damaging Bacterial Cells"

_applsci, doi:10.3390/app12199633_

Round 1

Reviewer 1 Report

The manuscript with the title "Mechanistic Exploration on Visible Light-Activated Car- 2 bon/TiO2 Hybrid Dots Damaging Bacterial Cells" is very well written. I have some minor suggestions:

could the authors please suggest some real-world applications? 

how do the authors conclude that the antibacterial effect is a result of C/TiO2 and not just the carbon dots? did they perform any experiment proofing that?

when the antibacterial effect is being measured, we should make sure that all the bacteria are getting killed as even a small number of them can grow rapidly, to use this technique, what will be the safe dosage of C/TiO2 to be used? and again in which industry we can use this technique?

Author Response

could the authors please suggest some real-world applications?

Response: As suggested, we have commented in Summary and Conclusions that these hybrid dots are more advantageous than the widely pursued dye-sensitized colloidal TiO2 for broad applications beyond antimicrobial uses, such as in photocatalysis for CO2 conversion and waste management.

how do the authors conclude that the antibacterial effect is a result of C/TiO2 and not just the carbon dots? did they perform any experiment proofing that?

Response: There could not be free carbon dots under the experimental conditions for the preparation and purification of the hybrid dots.

when the antibacterial effect is being measured, we should make sure that all the bacteria are getting killed as even a small number of them can grow rapidly, to use this technique, what will be the safe dosage of C/TiO2 to be used? and again in which industry we can use this technique?

Response: Good point, but that was not the case in this study, as the bacteria did not grow back in the post-treatment plating with incubation conditions designed for bacterial growth. On potential industrial uses, we see major opportunities in food processing industry, though this project is still in the fundamental research domain.

Reviewer 2 Report

 Audrey et al reported  Mechanistic Exploration on Visible Light-Activated Car-2 bon/TiO2 Hybrid Dots Damaging Bacterial Cells”  . I will consider publishing your current paper after minor revision. Although the subject of this manuscript is very interesting, the content a little disappointing, and doesn´t fit neither the Title nor the Abstract.

1. I had a close look and found the manuscript is partially hard to read and comprehend, but the manuscript should be rewritten more readable.

2. The authors need to compare their results with published articles.

3. The authors should provide some information or discussion on the composition or chemical structure of the used materials.

4. The Conclusion should provide a critical (!) assessment in comparison with the related papers, including limitations and advantages.

5. How the authors estimated the exact size of the synthesized nanoparticles?. some references should be added. Some of the related references are given as well:

https://doi.org/10.1016/j.cis.2020.102316

https://doi.org/10.1016/j.ijpharm.2020.120021

https://doi.org/10.1016/j.arabjc.2021.103323

https://doi.org/10.1016/j.microc.2020.105663

Author Response

  1. I had a close look and found the manuscript is partially hard to read and comprehend, but the manuscript should be rewritten more readable.

Response: Based on the topics of the papers suggested in the comment #5 below, it appears that the reviewer was probably more interested in mechanisms with respect to the hybrid dots as antimicrobial nanomaterials, but the reported work was on biological consequence/mechanism due to the action of the light-activated dots, as reflected in the title and abstract.

  1. The authors need to compare their results with published articles.

Response: We have not found any other studies of similarly structured carbon/TiO2 hybrid dots with respect to the biological consequence/mechanism due to their antibacterial activities.

  1. The authors should provide some information or discussion on the composition or chemical structure of the used materials.

Response: The information has been provided in our recent publications, which are cited appropriately in this manuscript. Only information not reported in the previous papers is included here to avoid unnecessary duplication.

  1. The Conclusion should provide a critical (!) assessment in comparison with the related papers, including limitations and advantages.
  2. How the authors estimated the exact size of the synthesized nanoparticles?. some references should be added. Some of the related references are given as well:

https://doi.org/10.1016/j.cis.2020.102316

https://doi.org/10.1016/j.ijpharm.2020.120021

https://doi.org/10.1016/j.arabjc.2021.103323

https://doi.org/10.1016/j.microc.2020.105663

Response: On both comments, we did go through the suggested papers carefully but found that the topics in those papers are too far from the focus of this work, which would make the suggested “comparisons” not only difficult but also very odd to other readers.

Reviewer 3 Report

Overall the study is well-conceived and the manuscript well-written. However, it requires some minor editing before it can be published. My thoughts are as follows:

- please provide the strain number and supplier name of B. subtilis

- please provide the supplier of Luria–Bertani (LB) agar

- The authors wrote „The final bacterial cell concentration in each well was about ~106 -107 CFU/mL, and the final dot concentration was varied as needed.” (Page 3, line 103). What does the author mean when he writes: the final dot concentration was varied as needed? Please specify the used concentrations. The author can also specify what they depended on. 

Author Response

- please provide the strain number and supplier name of B. subtilis

please provide the supplier of Luria–Bertani (LB) agar

Response: Both provided in the revised ms, as requested.

The supplier name of B. subtilis was added, but no strain number was provided by the supplier at the time of purchase. LB agar was purchased from Fisher Scientific.

- The authors wrote „The final bacterial cell concentration in each well was about ~106 -107 CFU/mL, and the final dot concentration was varied as needed.” (Page 3, line 103). What does the author mean when he writes: the final dot concentration was varied as needed? Please specify the used concentrations. The author can also specify what they depended on.

Response: Revised. “The final bacterial cell concentration in each well was about ~106 -107 CFU/mL, and the final dot concentration was varied in the range of 0.01 mg/ml to 0.1 mg/mL, depending on the needs of individual experiments. The actual dot concentrations used in given experiments were indicated in the figure captions.”